# Object Detection at Level Crossing Using Deep Learning

**DOI:** 10.3390/mi11121055

**Published:** 2020-11-29

**Authors:** Muhammad Asad Bilal Fayyaz, Christopher Johnson

**Affiliations:** Department of Engineering, Manchester Metropolitan University, Manchester M15 6BH, UK; c.i.johnson@mmu.ac.uk

**Keywords:** railway level crossing, sensing system, algorithms, deep learning

## Abstract

Multiple projects within the rail industry across different regions have been initiated to address the issue of over-population. These expansion plans and upgrade of technologies increases the number of intersections, junctions, and level crossings. A level crossing is where a railway line is crossed by a road or right of way on the level without the use of a tunnel or bridge. Level crossings still pose a significant risk to the public, which often leads to serious accidents between rail, road, and footpath users and the risk is dependent on their unpredictable behavior. For Great Britain, there were three fatalities and 385 near misses at level crossings in 2015–2016. Furthermore, in its annual safety report, the Rail Safety and Standards Board (RSSB) highlighted the risk of incidents at level crossings during 2016/17 with a further six fatalities at level crossings including four pedestrians and two road vehicles. The relevant authorities have suggested an upgrade of the existing sensing system and the integration of new novel technology at level crossings. The present work addresses this key issue and discusses the current sensing systems along with the relevant algorithms used for post-processing the information. The given information is adequate for a manual operator to make a decision or start an automated operational cycle. Traditional sensors have certain limitations and are often installed as a “single sensor”. The single sensor does not provide sufficient information; hence another sensor is required. The algorithms integrated with these sensing systems rely on the traditional approach, where background pixels are compared with new pixels. Such an approach is not effective in a dynamic and complex environment. The proposed model integrates deep learning technology with the current Vision system (e.g., CCTV to detect and localize an object at a level crossing). The proposed sensing system should be able to detect and localize particular objects (e.g., pedestrians, bicycles, and vehicles at level crossing areas.) The radar system is also discussed for a “two out of two” logic interlocking system in case of fail-mechanism. Different techniques to train a deep learning model are discussed along with their respective results. The model achieved an accuracy of about 88% from the MobileNet model for classification and a loss metric of 0.092 for object detection. Some related future work is also discussed.

## 1. Introduction

Different projects that directly address the issue of over-population and ever-increasing demand for train services, which our current system is unable to cope with, have been initialized within Great Britain. Some of these major project plans for Control Period 6 (CP6) are mentioned by the Network Rail [1]. Some key projects include the Cross Rail Project [2], which will allow 1.5 million more people to travel in Central London within 45 min. Another project is the Great North Rail Project [3], which is part of Britain’s Railway Upgrade Plan connecting different towns and cities. Forecasts in the Trans-European Transport Network program predict that Trans-European High Speed (HS) network (category I and II lines) will more than double to reach a length of 22,140 km long by 2020, compared to its length of 9693 km in 2008. By 2030, this network is expected to comprise of 30,750 km of track, and traffic will have risen to approximately 535 billion passengers per annum [4]. Such expansions within rail industries demand electrification of the existing systems and an upgrade of these technologies will become inevitable. The expansions ultimately increase the number of junctions, intersections, and level crossings (LCs). A level crossing is where a rail line is crossed by a right of way without the use of a tunnel or bridge. There are currently 7500–8000 operational level crossings within Great Britain [5] and about 7000 are actively used on Network Rail managed infrastructure. Out of these 7000 level crossings, around 1500 are on public vehicular roads and others on a public footpath or private road. With such numbers, the risk associated with level crossings is significantly high, where heavy machinery cross with high speeds, and road users such as vehicles and pedestrians misuse level crossings during its operational cycle.

Misuse from road users accounts for nearly 90% of the risk encountered in the previous five years. Level crossings account for nearly half of the potentially higher risk incidents on British railways. For Great Britain, there were three fatalities and 385 near misses at level crossings in 2015–2016. Furthermore, in its annual safety report, the Rail Safety and Standards Board (RSSB) highlighted the risk of incidents at level crossings during 2016/17 with a further six fatalities at level crossings including four pedestrians and two road vehicles. Level crossings account for 8% of the industry’s risk compared to other accidental risk factors within the UK’s rail network. In addition to these six major incidents, the RSSB recorded 77 minor injuries, 39 shock and trauma incidents due to near misses, and six collisions between train and road vehicles. The RSSB suggested that the major factors for most of these incidents were related to the design of the level crossing [6]. Furthermore, the ORR has suggested that approximately 73% of level crossing fatalities involved pedestrians [7].

Additionally, all technology employed in the UK network must satisfy the “Network Rail Assurance panel Process (NRAP)”, which governs rail processes. When a change is introduced that could affect the risk profile of the Network Rail infrastructure, NRAP ensures the compliance of the respective engineering processes and technologies with Network Rail’s responsibility, health, and safety systems. The Level Crossing Strategy and Risk Assessment is considered in detail within the NR/L1/XNG/100/02 and NR/L2/OPS/100 standards. Furthermore, technologies must be approved through stringent business regulations and standards (NR/L2/RSE/100/06) that have reached TRL8 or above. Risk components such as interlocking devices will be reviewed against the NR/L2/RSE/100/06 with approval weighed against the risk to the business. All aspects of any proposed system will be reviewed for redundancy, ensuring that even small changes to the software system ensure system fail-safe expectations [8].

To ensure a safer and reliable system at level crossings, the use of deep learning has been introduced. The deep learning algorithm is integrated with the existing sensing system, which further reduces the cost and justifies the risk model for new applications proposed at level crossings. The paper briefly discusses the existing sensing systems and algorithms used at level crossings and outlines why Closed Circuit TeleVision (CCTV) and radar are the preferred choices for level crossing application. CCTV is used as a primary sensor to detect obstacles at level crossings, whereas, radar is installed as a secondary system to add another layer of resilience for a “two out of two” logic channel. The results demonstrate the effectiveness of deep learning technology and its ability to detect and localize obstacles at level crossings.

The present work discusses why automating a level crossing’s operational cycle is important and gives a brief overview of existing sensors and their relevant algorithms. After preferring particular sensors, the proposed work used deep learning technology and integrated it with an existing system to classify and detect obstacles. The work demonstrates a novel application of deep learning at level crossings within the rail industry. The “two out of two” interlocking system, in particular, with deep learning has not been used within the rail industry specifically for level crossings. The present work concludes with some remarks and relevant future work.

## 2. Literature Review

Different types of level crossings are present within Great Britain depending on its location, traffic type, usage, and technology used. Some level crossings are automatic, where no railway staff are present and the whole operational cycle is automated, whereas, some level crossings are passive with no staff and warning signs while others have warning signs and railway staff. The present work was concerned with level crossings, where sensing systems are present. The proposed work was integrated with an existing sensing system to automate the whole operational cycle of level crossings without the need of railway staff. Traditionally, automatic level crossings depend on sequential operations, which automated the operational cycle, but was incapable of making intelligent decisions. The proposed work will detect and localize the obstacle present at level crossings and make intelligent decisions to continue the operational cycle or alarm the train driver to any danger.

It is essential to briefly discuss the state-of-the-art sensors present at level crossings before preferring one particular sensor for its application at level crossings. Once a particular sensor is selected, the results from deep learning are mentioned. The results demonstrate how a model can detect and localize an obstacle at the level crossing, which is essential for automating the process.

### 2.1. Sensing System and Algorithm

Traditional sensors are installed either inside or on rail lines, which are called intrusive sensors. Intrusive sensors are costly and disrupt the rail system during their installation and maintenance, which further increases the cost and shortens the product lifecycle. Intrusive sensors were replaced with non-intrusive sensors, which are installed outside rail lines and do not affect the rail system during installation and maintenance. Therefore, the product is comparatively cheap with a longer product-lifecycle.

Some post-processing techniques are required to process the information acquired from any of the above-mentioned sensors to analyze and differentiate between obstacles present at the level crossing. Most of these traditional algorithms detect the foreground by comparing pixel values with background pixels. However, the environment at level crossings is complex and dynamic with growing vegetation and the presence of many harmless objects.

#### 2.1.1. Sensing Systems

Some of the earliest proposed intrusive sensors were inductive loops [9], whose working principle depends on inducing the electric circuit. However, new vehicles are made of composite materials whose detectability is significantly reduced. Other intrusive sensors include strain-gauge or piezometers [10], whose functionality depends on the deformation caused by the pressure of an obstacle. Such sensors were unable to detect small children, hence these sensors were replaced by many non-intrusive sensors.

Some common non-intrusive sensors include stereo cameras [11], where two cameras are required to obtain a 3D representation of the level crossing. Different post-processing techniques are required to analyze the information, which often leads to false-positives [12]. Thermal cameras are another alternative, and their working principle is based on temperature change, but some inert bodies or bodies with the same temperature as the background remain undetected [13]. Ultrasonic [14], lasers [15], and LiDAR [16] work on the same principle of emitting a pulse and receiving the reflected pulse from an obstacle for detection. Such approaches do not work effectively in adverse weather conditions or require multiple sensors to cover an entire region of the level crossing, thus increasing the cost of installation. The inability of these mentioned sensors to work effectively in adverse weather conditions requires a secondary sensing system to add a layer of resilience to the system.

Currently, the most commonly used sensors at level crossings are radar and CCTV. CCTV provides constant pixel values of the scene at the level crossing, which is fed to the signaler or manual operator to decide if any danger is present at the level crossing or to start the automated process. The functionality of CCTV cameras is reduced in low light conditions, however, a small lighthouse is often built next to the CCTV housing box to provide sufficient light at night [9]. Works in [17] discuss the relevant tests (e.g., hardware and function test required before verifying and implementing CCTV at the level crossing area). These tests are an essential part of risk management. Authors in [18] discussed radar at four-quadrant gate level crossings. Radar is often the primary choice of a detector at level crossings because of its low cost, high product-lifecycle, and low maintenance. The radar must be installed at a certain distance from other radars in the surrounding area. The 24 GHz radar takes about 8 s to warm up before scanning the area at the level crossing at a rate of 1 scan per second. If three successive scans are cleared, the automated process of closing the barriers at the level crossing is initiated [19]. Table 1 provides a brief summary of each obstacle installed or can potentially be installed at level crossings.

#### 2.1.2. Algorithms

Traditional algorithms use Gaussian probability function (e.g., single [20], multiple [21], or mixture of multiple [22] Gaussian to either compare pixel values with a pre-defined threshold value or adjust with the noise and any outliers, and classify according to their pixel values. The authors in [20,23] discussed the use of probability density function using K-estimators, which was compared with the pre-defined threshold value. Other sophisticated methods rely on linear functions (e.g., subspace learning [24]), where the algorithms learn information, variance [25], and correlations [26] between pixel values and use a linear function to separate the foreground from background pixels. Some algorithms used linear regression to calculate the intensity of pixels using the support vector regression method [27], while others are based entirely on median values to the model background such as temporal median filter (TMF) [28]. Most recently, basic machine learning algorithms such as unsupervised learning or support vector machine, which uses a hyperplane in high dimension to separate the data with outliers [29]. All these mentioned algorithms rely on raw pixel values to calculate the mean, variance, or median to model the background or just compare the background with new pixels using a pre-defined threshold to detect an object.

Traditional sensors and their associated algorithms did not provide sufficient information to analyze the data for classification and detection of obstacles at level crossings. To ensure a safer sensing system, the proposed algorithm uses deep learning technology to integrate it with the existing sensing system [30]. Deep learning, which is a subset of machine learning, is composed of multiple processing layers. These multiple layers learn representations from the large dataset using the backpropagation algorithm, which indicates how a machine should change its internal parameters. These internal hyperparameters are used to compute the representation of a layer from the representation of the previous layer. With enough representation learned, very complex functions are learned and used for applications such as classification and detection. For classification tasks, the higher layers of representations increase the aspects that are required for discriminating the objects and suppressing the irrelevant variations [31]. The ability to learn itself and classify the objects without the intervention of a manual operator makes the use of deep learning the most appropriate choice for level crossing applications. The works in [32,33,34] mention details about the training and computation power. For a detailed review on the existing sensing system and their relevant algorithms, please refer to the work in [35].

### 2.2. Methodology

The ultimate aim of this research was to make the operational cycle at a level crossing safer and more reliable with the use of new technology and innovative methods. The proposed method should justify the cost with the associated risk and provide sufficient data for risk analysis, where relevant authorities can take more effective precautionary measures.

To select the most appropriate sensing system, a survey on the available sensor was presented earlier in this paper. These sensors are either used or has the potential for its applicability at level crossings. The limitations and preference for each sensor are discussed and summarized in Table 1. The best possible combination of CCTV and radar is proposed. The primary choice of the proposed system is CCTV, which is used and integrated with deep learning techniques to detect and localize objects. The radar system is used as a complementary sensor to add resilience to the sensing system. Radar systems are already installed at most level crossing areas, and they are capable of detecting small objects regardless of adverse weather conditions. However, the traditional methods integrated with a radar system use raw data to calculate speed and direction, etc. The raw data do not provide sufficient information to classify objects (e.g., cannot differentiate between an inanimate small object and a child) with risk. Another work proposed the idea of integrating CNN with radar data using different post-processing techniques, for example, some have used 3D cube data to classify objects [36], while others have used Vision and LiDAR to label data using Frequency Modulated Continuous Wave (FMCW) radar [37]. These proposed methods classify objects for road users and have not been utilized in the rail industry; moreover, the radar used is not same as that installed at railway level crossing sites, which further increases the cost of the system implemented. For this work, the model primarily relies on the Vision system integrated with deep learning and uses radar to detect objects as a secondary sensing system in the “2oo2” logic model. Figure 1 presents the interlocking system proposed with “two out of two-2oo2” logic channels, where a comparator is used to compare the executed output from both channels. The radar system detects objects and compares them with the given classification of an object using CCTV. If the two outputs show an obstacle present at the level crossing, then the operational cycle at the level crossing is initiated if it is clear to do so, or the train driver is alerted. If, however, the processing channel fails, then a fail-mechanism is initiated and system is shut-down to alert the train driver in time. Figure 1 demonstrates the “2oo2” logic interlocking system.

As shown in Figure 1, the output from the Vision system using deep learning and radar system are compared for the vote. If each channel gives an output that shows an obstacle is present, the vote would be to alert the train driver. If both channels process output and does not show any obstacle present at the level crossing, the whole process of the operational cycle will be continued, automating the process. However, if one channel fails to give output or in the case of “fail-system”, the system will alert the train driver about the situation. Possibly, the information from other channels could be transmitted in real-time using GSM-R, so the train driver can decide in the given time.

The sensors require some post-processing techniques to analyze the data and make predictions where necessary or facilitate a manual operator in the detection of obstacles at level crossings. The traditional and manual algorithms mostly rely on the concept of subtracting pixels from background pixels to detect foreground. The ineffectiveness of this approach and its limitations are discussed and compared with the deep learning technology, which is proposed for the application. Deep learning technology can learn features automatically to classify and detect an obstacle and does not rely on pixel values.

Different deep learning techniques and models are used for classification and detection depending on its application and availability of the data. These techniques use the pre-labeled dataset to train the neural network (NN) for image classification. Two different methods are available to design and train the neural network; designing and learning from scratch or using transfer learning. The proposed work will use both of these methods to compare the results and analyze which particular method is suitable for application at level crossings.

The adapted techniques and approaches for deep learning are as follows:A neural network is designed and trained from scratch using a custom dataset downloaded from an open-source (ImageNet) for the classification of objects at level crossings.Image classification results obtained from a model trained from scratch are compared with the neural network trained using transfer learning techniques using pre-trained models.Train object detectors that provide results (e.g., classification, localization, and detection for multiple objects at level crossings.Discussion on risk assessment for the given sensing system for its installation and maintenance.

The results from these NNs were evaluated using a “dataset”, which was unseen data to the NN during its training phase. The test data provides sufficient information about the NN performance and accuracy. These given metrics will help to select the most appropriate technique within deep learning for the Vision system and radar.3. Results and Discussion

CCTV and radar are the preferred choices of sensors because of their low cost and maintenance, and their ability to provide sufficient information for deep learning to learn for its applicability at level crossings. Deep learning is a proposed technology integrated with these sensors. Deep learning models require labelled images to learn representations for classification and detection. Labeled images for classification are available from ImageNet and for detection, images are available from the “Open Image Dataset”. For classification, the images contain relevant labels, whereas the images for detection also contain coordinates for bounding boxes. For classification, the model is trained from scratch to visualize the effect of certain layers and representations learned during the training process. Once trained, the dataset is used in transfer learning; a method to utilize pre-trained models, which are trained on millions of images for thousands of categories. Transfer learning techniques achieve higher accuracy from a small given dataset since it uses general representations, features, and motifs from a pre-trained model.

## 3. Results and Discussion

### 3.1. Dataset

The ImageNet dataset is an opensource platform (ImageNet, 2020), which provides over 15 million high-resolution labelled images with around 22 categories for classification. The required application for Level Crossing should classify and detect objects particularly pedestrian (adult and child), bicycles and vehicles. Therefore the dataset downloaded from the ImageNet should be of these particular categories. These images are arranged in tree-directory format, where folder name represents the label for images within that particular folder. These images are divided into “train” and “validation” dataset. The training dataset is used for training the Neural Network and Validation dataset is used to evaluate the performance of such Networks during the training. The Validation helps the practitioner to avoid Overfitting. Sometime another dataset called “test” is finally used to evaluate the training and validation processes. Accuracy and Losses obtained from such dataset is used as an evaluation metrics to compare results for each given model. Often the downloaded images from open-source are not very specific to their given category and represents another object with more dominant features. For example, the dataset downloaded for pedestrian (adult and child) contains vehicles or bicycles within major part of an image. Such images disrupts the feature learning process and makes the model ineffective. Therefore, the downloaded images are manually checked and any image with strong dominance of another objects are excluded from the dataset. To compensate the number of images, the images from Level Crossing at local site are collected and added to the dataset. However, the dataset may still contain some images where an image represents another object and small error is still expected. Excluding images from the dataset may reduce the diversity of the given categories. To overcome the issue of diversity, data augmentation techniques are used to add images to the dataset as shown in Figure 2. The images in Figure 2 represents images after certain operations e.g., rotation, shift, zoom and flip are applied to particular image.

CCTV is the preferred choices of sensors because of their low cost and maintenance, and their ability to provide sufficient information for deep learning to learn for its applicability at level crossings. Deep learning is a proposed technology integrated with these sensors. Deep learning models require labelled images to learn representations for classification and detection. Labeled images for classification are available from ImageNet as shown in Figure 3 and for detection, images are available from the “Open Image Dataset”. For classification, the images contain relevant labels, whereas the images for detection also contain coordinates for bounding boxes. 

For classification, the model is trained from scratch to visualize the effect of certain layers and representations learned during the training process. Once trained, the dataset is used in transfer learning; a method to utilize pre-trained models, which are trained on millions of images for thousands of categories. Transfer learning techniques achieve higher accuracy from a small given dataset since it uses general representations, features, and motifs from a pre-trained model for classification and detection.

### 3.2. Classification

A simple neural network was trained to further add convolution layers to demonstrate the effect of convolution and visualize the representation learned from the given dataset.

Table 2 includes the trainable parameters, a summary of the architecture, and accuracy achieved during the training process. The difference in accuracy as due to “overfitting”, a common problem where the model has learned sufficient representation on the training dataset but not enough to classify a new unseen image. To overcome the “overfitting” problem, the model should be fed with more data. Often data are not available, hence data augmentation techniques (e.g., rotation, resize, flip or shear) are applied to an image to add more diversity. Another technique is to use the “drop out” layer, which randomly drops learned features, thus reducing the number of trainable parameters and improving the accuracy. The final accuracy on the validation dataset (which was used as a benchmark for the models’ accuracy) increased from 41.75% to 69.63%. To better understand the effect of the convolution layer, often the visualization of such representations during the training process is used, as shown in Figure 4.

In the first layer, the activation from the neural network has retained all the information from the given image (e.g., bus or car). It also contains some edges or collection of apparent edges that is interpretable by human eyes. The activations or representations become more complex in deeper layers of the neural network as they are more abstract and less interpretable. These complex features correspond to the specific class rather than a visual interpretation of the generic object. The black boxes in the subsequent layers of the neural network demonstrate the sparsity of the network. These black boxes represent that the pattern encoded by the particular filter is not found in this particular location. The convolution neural network is fed with raw data containing specific objects, and the subsequent layers within the CNN learn relevant information specific to the class and leaves irrelevant information (e.g., the visual appearance of the image). For this reason, any object similar to the given object in the image, regardless of its orientation and size, will be predicted accurately at the level crossing. This is impossible with the traditional algorithms, which relied on background pixel values rather than the features.

The accuracy of these models was further improved using transfer learning techniques on pre-trained models. Details of these models and their accuracy are mentioned in Table 3.

### 3.3. Detection

Often, the relevant authorities are not interested in classification only, rather requiring precise location of these detected objects (e.g., object detection). Object detection models require more information in the given dataset (e.g., bounding boxes and labels for the given categories). Different models are used with transfer learning techniques, and their results were compared using the COCO evaluation metrics. Details of these models and their loss metrics are mentioned in Table 4.

The results for classification and detection demonstrate the effectiveness of the model, where the given model classify and detect the required obstacle in real-time. Real-time detection with an accuracy that exceeds human error-rate is essential to automate the level crossing operational cycle using intelligent-decisions. Since the integrated technology does not incur additional costs and utilizes the existing sensing system, the installation cost is significantly reduced and risk assessment is easily managed and processed.

The results for Image Detection are finally evaluated with real test-dataset, which contains images from local sites.

The Figure 5 demostrate the effectiveness of the model, since it can classify and detect object within dynamic environment of Level Crossing with high accuracy. The model evaluation at real-site scenes using images obtained from Camera shows its adaptiability at Level Crossing site, where Deep Learning technology is integrated with CCTV. An essential part for application and implementation of new proposed solution is the Risk Management, which is briefly discussed below.

## 4. Risk Management

The Risk Assessment Method Statements (RAMS) systems is an enlarged engineering discipline, which originated from the concept of safety and reliability [48,49]. This was introduced to assess the product failure and human error, where the first assessment techniques were established in the 1940s in the “Failure Mode and Effect Analysis (FMEA)”. The FMEA was further evolved to “FMECA”, where the aspect of criticality was also introduced [50]. For specific risk assessment, the FMECA technique is used for its flexibility in risk assessment stages: Identification, Analysis, and Evaluation. As mentioned earlier, the FMECA should address six fundamental questions for each possible failure mode. Below is a discussion of every possible failure point during the training, processing, and within the whole system.

In the design process, the system may have several issues that need to be addressed and resolved. The data used for training may not be sufficient or may not be from the same distribution (e.g., level crossings). The same distribution data are required to avoid any bias in the network, likewise, enough data should be available to learn sufficient representation for classification and detection. To avoid any bias and underfitting, two different techniques were utilized for this work. First, two different datasets were downloaded from ImageNet for models trained from scratch. This will provide sufficient data to learn representation. Second, transfer learning techniques were used where pre-trained models (trained on millions of images) were used for our application. This will allow the neural network to retain the abstract features learned from these millions of images and utilize them to create higher motifs and groups of motifs for a particular category. For the radar dataset, the MATLAB functions were used to simulate a sufficient dataset using different pre-defined scenarios. Such a dataset will provide diversity and enough representation to learn from the neural network avoiding bias and underfitting.

Creating a e from scratch is an art and requires time and careful engineering to design the architecture. Poorly designed architecture will fail to classify obstacles accurately and will unnecessarily create deep networks without increasing efficiency. To avoid such instances, the architecture of pre-trained neural network was used. These architectures have been trained and evaluated on different applications and datasets. A similar type of neural network will be used for training on the radar dataset since the radar dataset is new and has not been widely trained. A simple neural network was used in its initial stage of progression.

The training of this neural network requires high computational power for efficient training. The high computational power can be achieved from GPUs, hence risk associated with hardware is also considered during the processing phase of the system. The system it is trained on must have an updated risk assessment for all its wiring and environment. Any failure within the system will affect the training and ultimately the whole system. The risk assessment of these systems was updated and monitored to ensure no failure modes were expected during the training and optimizing of the neural network.

Once trained and tested, the final neural network file was updated to the level crossing’s installed system at the site. These systems are integrated with the sensors, where a continuous stream of data from CCTV and radar are fed to perform classification and detection. These classification and detections were used to automate the operational cycle of the level crossing and also collect the data for further risk assessment. To ensure the system worked properly, relevant risk assessments were performed for the whole system. For example, the system should be regularly monitored and updated and it should be installed in a safe component not accessible by the public and should be protected from vandalism. The sensors should be working properly along with the barriers, and any electrical component used to automate the operational cycle. Before implementation at a real site, the system should be tested at a remote physical site to ensure the safety and reliability of the system. Several test scenarios should be considered, which should replicate the level crossing site to test and evaluate the system in a complex environment. These test plans and techniques were used to ensure the whole system was working properly before its final implementation at the level crossing site.

### Other Applications

The same “model system” discussed earlier in this work for the efficient operation of level crossings is also applicable for other applications within the railway industry. The same trained model can be integrated with any Vision system within the rail industry. Some key areas within the rail industry, where the same model can be integrated, are discussed.

#### Classification of Passengers at Platform

Often, relevant authorities require statistics of specific categories using the railway platform at a specific time of the service. These statistics should include the passengers with bicycles or passengers with luggage or disabled chair. The data can help relevant authorities to analyze their traffic at a particular platform and accordingly upgrade the given services or facilities. The statistics will also allow the relevant authorities to analyze what particular platform is most busy and at what particular hour. The acquired information is fed to a simple yet powerful data visualization software (e.g., tableau, which gathers these data and visualize them in relevant graphs, providing information more clearly and efficiently.

#### Tracking Any Suspicious Behavior

Relevant authorities are always managing the traffic at the platform or within the train to detect and respond to suspicious behavior (e.g., suicidal attempt or violence). Busy traffic at the platform often makes it impossible for the manual operator to detect such activity in time or respond to it efficiently. To detect such occurrences based on certain instances (e.g., random movement of a person near rail lines or people gathering at one place), these instances are easily trackable using the deep learning model, which track any random or unwanted movements in the pre-defined area. The threshold value is used to alert the system if the detection is stronger than the given threshold value. This will give enough time for manual operators at the platform to respond and avoid any unwanted situations. The model can track such objects at the platform, which makes it easy for the relevant authorities to respond in a short timeframe.

These are some applications, which are achievable using the same deep learning model by using different pre-trained models available, along with their relevant weight files, to train models for a given application at level crossings. Other applications may include a census of passengers at the platform, managing passengers at the platform at the time of train arrival, or tracking objects at the private accessed site at the platform, etc.

## 5. Conclusions

The present work addresses the alarming threats associated with level crossings because of the on-going projects, expansion plans, and increasing number of level crossings throughout Great Britain and Europe. The accidents, fatalities, and near misses at the level crossing are mostly the result of misuse and irresponsible behavior of pedestrian at the level crossing. Therefore, the need for a novel solution to automate the operational cycle at the level crossing, which is capable of making intelligent decisions.

To choose a particular sensor, the present work provides a brief survey on traditional sensors that were installed on or inside rail lines (e.g., inductive loops and strain gauge). Such sensors disrupt the rail lines during its installation and maintenance, making the system costly and inefficient for its applicability at level crossings. An alternative to intrusive sensors is non-intrusive sensors. Non-intrusive sensors (e.g., radar and CCTV) are installed outside rail lines and do not disrupt the rail lines during installation and maintenance periods. Low cost and high product-lifecycle of these sensors make radar and CCTV the preferred choice for applications at level crossings. Radar and CCTV were installed as the primary and secondary sensing system, respectively, to add another layer of resilience and operate as a “2oo2” logic channel interlocking system. The use of deep learning with CCTV is a novel approach within the rail industry specifically at level crossings. Additionally, the use of the “2oo2” logic channel interlocking system is a novel application at level crossings.

Each of these sensors required some post-processing algorithms, which mostly relied on a predefined threshold value or finding mean, variance, or median to model the background scene. Such approaches are inefficient for the complex and dynamic environment of level crossings, where growing vegetation and harmless objects are often present. Therefore, the new proposed model should not rely on raw pixel values, but rather on the features and actual representation of the object. Therefore, the proposed work adapted deep learning technology to learn representations from the given object using multiple layers. These representations are used to classify objects or localize their precise location at the given frame. Such an ability to classify and detect obstacles allows relevant authorities to make intelligent-decisions and automate the operational cycle at level crossings.

To elaborate on the effectiveness and efficiency of the model, the results for classification using a model from scratch or trained using transfer learning techniques were discussed. The models trained for image classification achieved an accuracy of about 88% using pre-trained MobileNet model. For object detection, the MobileNet model achieved a loss metric of 0.092 on COCO evaluation metrics. The mentioned metrics suggest the efficiency of the given model for its applicability at level crossings. These models were trained and inferred with the existing sensing system, which further justifies the cost of the overall safety system at a level crossing. The ability of the given model to detect and localize obstacles without the interference of a manual operator will allow relevant authorities to automate the process more efficiently, hence reducing the associated risk at level crossings.

Finally, the proposed work discussed the risk assessment in detail using “FMECA” techniques. The risks associated with data collection, training the neural network, and deploying the network with existing sensing systems are discussed, which will help relevant authorities to analyze and manage the installation of the proposed system at level crossings.

## 6. Future Work

To improve the accuracy of the model, the dataset should be collected from the same distribution as that of level crossings. Relevant authorities should cooperate to collect the dataset purely from level crossings across different regions within Great Britain. This will remove bias from the dataset and improve the model’s accuracy. The same model can be used in different applications (e.g., census of passengers according to its classification, tracking any suspicious behavior, or alarming passengers across the red alert zone at the platform).

The deep learning model can be integrated with radar to classify obstacles, which was very difficult with traditional post-processing techniques. Traditional techniques use the information to calculate speed, range, and direction; however, the deep learning model can train on micro-doppler signals from radar to learn representations sufficient for classification. The micro-Doppler signals are unique for each given object at the level crossing and do not require a new radar model for its integration with deep learning.

## Figures and Tables

**Figure 1 micromachines-11-01055-f001:**
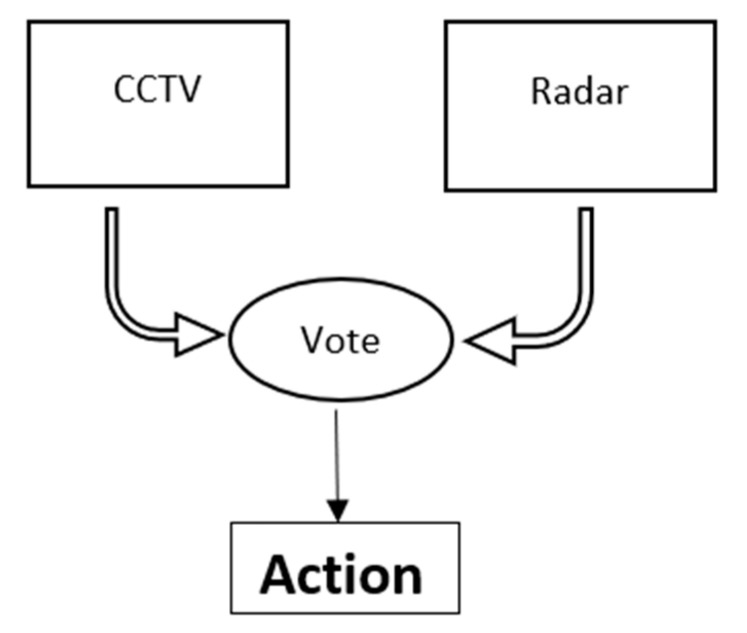
The ‘two out of two” logic interlocking system, where an output is executed if both channels agree, otherwise the fail-safe mechanism is executed.

**Figure 2 micromachines-11-01055-f002:**
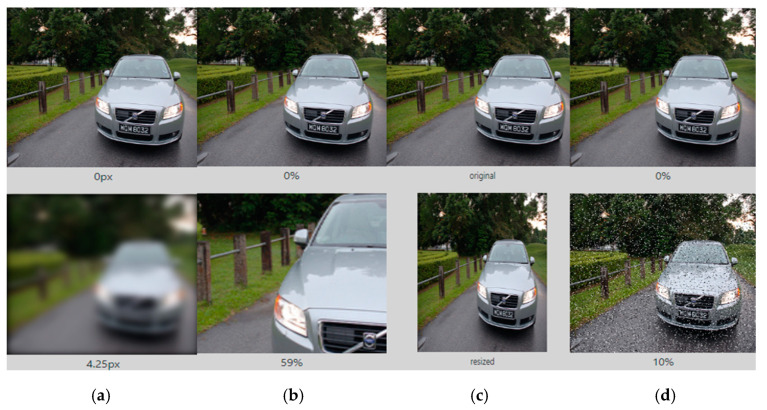
Different Augmentation and Pre-processing operation are applied to the dataset. (**a**) Blur of intensity 4.25 px (**b**) Crop of 59% (**c**) Resize (**d**) Noise addition of 10%.

**Figure 3 micromachines-11-01055-f003:**
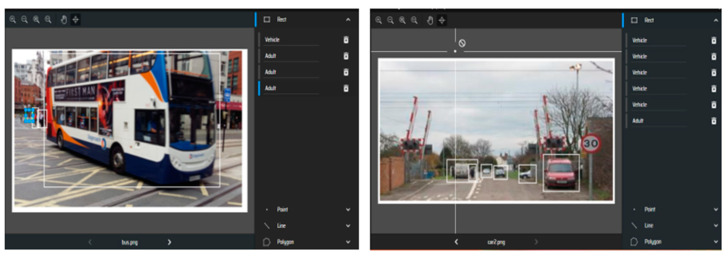
Annotated images where relevant categories are defined using bounding boxes. These bounding boxes coordinates are used to automatically generate a file required for training the model.

**Figure 4 micromachines-11-01055-f004:**
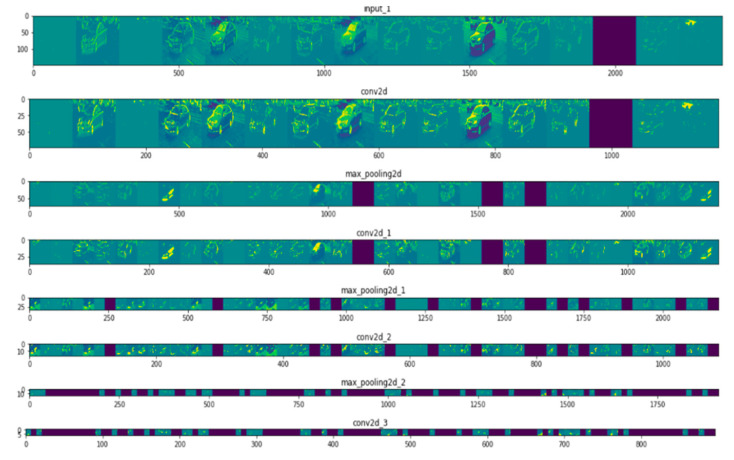
Visual representation of an obstacle (e.g., car through subsequent layers of convolutional layer during the training process).

**Figure 5 micromachines-11-01055-f005:**
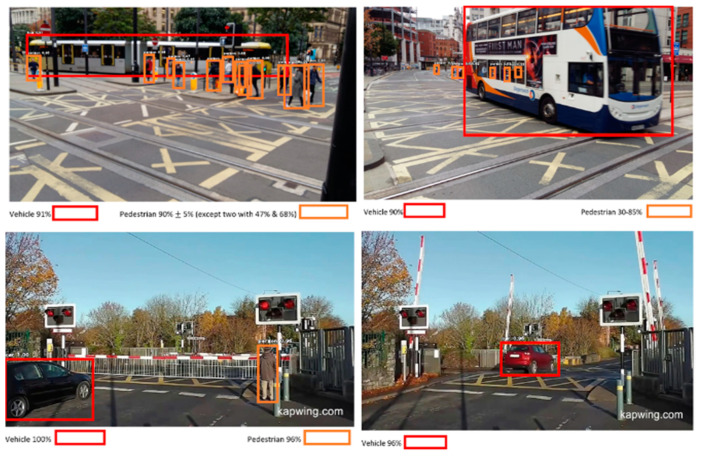
Object detection of multiple objects at two different sites of Level Crossing.

**Table 1 micromachines-11-01055-t001:** A brief comparison of different sensors available at level crossings, where five represents the worst choice and one represents the best choice for the given parameter.

Sensor	Equipment Cost	Maintenance	Product Life Cycle	False Positive	Impact on Weather
Inductive Loops	5	5	4	4	4
Strain Gauge/Piezometers	5	5	4	4	3
CCTV	2	3	2	3	3
Stereo Camera	3	4	2	3	3
Thermal Camera	4	2	2	2	2
SONAR	2	4	3	3	2
Millimeter-Wavelength Beam Interference	4	4	1	3	4
Laser Range Finder Beam (LRBF)	3	4	2	5	4
Light Detection and Ranging (LiDAR)	5	4	2	2	2
Radar	3	2	1	1	2

**Table 2 micromachines-11-01055-t002:** A simple neural network compared with the convolutional neural network (CNN). The CNN with a small increase in trainable parameters increased the accuracy on the training dataset from 41.75% to 69.63%.

Model	Architecture	Trainable Parameters	Accuracy
Traditional Neural Network	Input Layer	34M	Training acc: 53.84%
	Flatten Layer		Validation acc: 41.75%
	Dense Layer (2×)		
Convolutional Neural Network	Input Layer	44M	Training acc: 99.47%
	Convolutional Layer		Validation acc: 52.05%
	Max Pooling Layer		
	Flatten Layer		
	Dense Layer (2×)		
Convolutional Neural Network	Input Layer	3M	Training acc: 66.78%
	Convolution Layer & Max Pooling Layer (4×)		
	Flatten Layer		
	Dense Layer		Validation acc: 69.63%
	Dropout Layer		
	Dense Layer		

**Table 3 micromachines-11-01055-t003:** Brief information of the given model used for transfer learning and their associated accuracy achieved.

Model	No. of Trainable Parameters (Architecture with No Top)	No. of Trainable Parameters (Architecture with New Layers, e.g., Flatten Layer and Dense Layer)	Accuracy Training acc/Validation acc
VGG-16 [38]	14M	8M	74.67%/79.00%
VGG-19 [39]	20M	8M	72.83%/76.00%
DenseNet [40]	7M	16M	79.55%/87.50%
Inception [41]	21M	38.5M	83.17%/86.25%
Xception [42]	20.8M	52.4M	80.23%/85.25%
ResNet [43]	23.5M	52.4M	44.98%/53.00%
MobileNet [44]	3M	16M	86.02%/88.00%

**Table 4 micromachines-11-01055-t004:** Results from the given model using transfer learning techniques and their results are compared using the COCO evaluation metrics.

Model	Loss (COCO Evaluation Metrics)	Average Step Time (s)
Efficient-D7 [45]	1.15	0.161
RCNN [46]	2.178	0.315
ResNet [47]	0.721	0.133
MobileNet [44]	0.092	0.255

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
