# Peer review of "Object Detection at Level Crossing Using Deep Learning"

_micromachines, 2020, doi:10.3390/mi11121055_

Round 1
Reviewer 1 Report
The work proposed to use deep learning with CCTV and RADAR for safety control on real-way level crossing. While this use case is valid, I didn't find any novelty or detailed technical challenge discussions throughout the paper. The models and dataset used in the experiments were public, and the descriptions were general and already known.
I would suggest the authors to significantly rewrite the paper, redo the experiments with datasets collected on the real environments, and carefully analyse the safety implications. Besides the overall accuracy, what are the false negative and false positive ratios? If the behaviour of the deep learning system cannot be fully controlled, the setup may not be able to deployed in a real environment.
Reviewer 2 Report
Abstract:
- Don't start a new paragraph between lines 17 and 18
- "Make intelligent decisions" sounds like an intelligent agent is programmed to act in the world. Should it not rather be "inform human decision-makers accurately"?
- How can "traditional sensors..." be "implemented as a single sensor"? rather something like "traditional techniques are based on a single sensor"
- Don't explain what deep learning is in the abstract, expect that the reader is comfortable with these concepts
Introduction:
- The first sentence "Multiple projects are being planned" is really weird. Projects about what?
- The economic motivation is good
- RADAR can be replaced with "radar" throughout this paper, that would be my preference
- Can you make it more clear (here and in the abstract) which types of objects you are interested in?
(general comments on) Literature Review:
- It is weird to have one subsection "Sensing System & Algorithms" in your lit review, rather just a flat section
- I would blend some of this related work (especially the part about table 1) and some of the results & discussion (parts about table 2) into a separate section called "method"
(some missing items from) Literature Review:
- "Classification of railway level crossing barrier and light signalling system using YOLOv3" by Sikora et al
- https://www.railway-technology.com/contractors/infrastructure/corridor-ai/attachment/level-crossing-video-showing-machine-learning-detections/
- for a dual (deep) camera-radar object detection system: "Distant Vehicle Detection Using Radar and Vision" by Chadwick and Newman
- for a radar-only multi-object detection system (cars, people, bikes, etc): "RSS-Net: Weakly-Supervised Multi-Class Semantic Segmentation with FMCW Radar" by Kaul et al
Results and Discussion:
- see above comment about structural change and a "method" section above
- where is it made clear which sensor(s) you ended up using? just a camera or also radar?
- Figure 1 is pretty useless - I don't see how it helps me to view the layer activations
- Table 3 and Table 4 are what I wanted to see! Good numerical figures. Make sure you say which architecture you choose based on these figures.
Risk Management:
- I don't understand the (scientific) point of this section. Is it about the false positives/inaccuracy of the learned models and what effect that has in this application? If so, state some numbers here.
Round 2
Reviewer 1 Report
Since there is no much novelty in the techniques used, I would like to see substantial experiment results to support the application use case.
How would the RADAR provide a fail-safe solution for the CCTV system? I didn't see any experiment to support this.
How would the system behave in heavy rain or bad weather conditions? These are critical if we really want to deploy the system to replace manual control. In current experiments, the models and datasets are public and general, and it is not convincing that directly using them in the rail crossing application can straightforwardly achieve very good results.
Author Response
Hi,
Thank you for taking the time to review my paper and providing feedback. Please allow me to answer your questions.
I did not mention in my paper that my novelty is "Deep Learning" itself rather the application of it within Rail Industry particularly at Level Crossing. Not a single Level Crossing has yet used Deep Learning to classify or detect objects and still depend on manual operators hence prone to human error, therefore the novelty is still the application of Deep Learning at Level Crossing.
The present work mentions results from more than one notable model and experimented with datasets to demonstrate its effectiveness for its specific applicability at Level Crossing. The model used for Transfer Learning is public but their use and training are very specific and require careful consideration of hyperparameters for its optimization during the training process. There is also an on-going debate about the novelty in Deep Learning, if I add one layer to e.g. MobileNet architecture does it become a new model altogether? That is another topic for another paper but for my paper, I used the public models to train specifically for my application. I have also trained on a CNN from scratch with my own architecture. Therefore, I would say the novelty of CNN is present at work as well.
Thank you for mentioning about the Radar part, maybe I needed to make it more obvious. Please find the changes (highlighted in red). The Radar is used as usual at Level Crossing and only their output is used for the 2oo2 logic channel Interlocking system. The technical aspect of Radar was out of scope for my present paper. However, I did try to make it clear.
Hope you can reconsider your comments and accept my answers and highlighted changes in the paper. Hope to hear from you soon.
Reviewer 2 Report
I am not outright rejecting this paper because the application is interesting/important and has obvious benefit and I believe that there might be a useful contribution here, however:
- I do not see any contribution to theory (CNN object detection is well studied).
- The written English is difficult to navigate (changes required are structural, and beyond copy-editing)
- In particular, CCTV+radar sensor combination is "proposed" but I do not see any results with radar included
Overall, it is very unclear what the key "idea" or "contribution" is.
Author Response
Hi,
Thank you for taking the time to review my paper and providing feedback. Please allow me to answer your questions.
I did not mention in my paper that my novelty is "Deep Learning" itself rather the application of it within Rail Industry particularly at Level Crossing. Not a single Level Crossing has yet used Deep Learning to classify or detect objects and still depend on manual operators hence prone to human error, therefore the novelty is still the application of Deep Learning at Level Crossing.
The present work mentions results from more than one notable model and experimented with datasets to demonstrate its effectiveness for its specific applicability at Level Crossing. have also trained on a CNN from scratch with my own architecture. Therefore, I would say the novelty of CNN is present at work as well.
Thank you for mentioning about the Radar part, maybe I needed to make it more obvious. Please find the changes (highlighted in red). The Radar is used as usual at Level Crossing and only their output is used for the 2oo2 logic channel Interlocking system. The technical aspect of Radar was out of scope for my present paper. However, I did try to make it clear.
Hope you can reconsider your comments and accept my answers and highlighted changes in the paper. Hope to hear from you soon.
Round 3
Reviewer 1 Report
This version has been improved compared with previous versions. The authors needs to check carefully about the text editing. For example, Figure 1, the caption is several lines below the Figure itself.
Author Response
Hi,
Thank you for taking the time to review my paper and asking for minor revisions. I have corrected some grammatical errors in the manuscripts.
However, the manuscript I submitted has Figure 1 starts from line 221 and had only one line gap. I removed that one line as well between caption and text. If you meant some other figure in the text let me know, please. I did however checked every figure and they are without a gap in the submitted manuscript.
Thank you for your time hope the minor issue you mentioned is addressed in the given manuscript.
Reviewer 2 Report
I have two main comments, related to radar.
First, on line 198 you mention that "cannot differentiate between an inanimate small object and a child...". Object-level discrimination with certain types of radar are indeed possible, so please amend this statement and e.g. add references to examples:
- Palffy, Andras, et al. "CNN based Road User Detection using the 3D Radar Cube." IEEE Robotics and Automation Letters 5.2 (2020): 1263-1270.
- P. Kaul, D. De Martini, M. Gadd, and P. Newman, “RSS-Net: Weakly-Supervised Multi-Class Semantic Segmentation with FMCW Radar,” in Proceedings of the IEEE Intelligent Vehicles Symposium (IV), 2020
The first, Palffy et al, discriminates between bicycles and people. The second, Kaul et al, goes further and discriminates between cars, people, bicycles, poles, etc. The second, Kaul et al, also fuses camera and radar as you do in your work.
Second, thank you for providing the figure that shows radar and camera combined by a vote. Please detail further in the text how this vote works ("vote" is in the figure, but never in the text). Additionally, make it clear whether radar detections are used raw or whether this submodule is also trained as a machine learned model, how they are associated with camera locations (extrinsic calibration etc).
Author Response
Hi,
I hope you are doing well and thank you for taking the time to review and highlight the minor revisions.
I have tried to answer the questions in the given manuscript and highlighted in green for easy accessibility. Please find the reasons below as an attempt to answer your reviews.
First Review:
My novelty of the work is not a new proposed system rather its application at Level Crossing. It has not been done before within Railway and has benefits compared with traditional sensing systems. I did not focus on the Radar part and it was not my primary focus. Line 204-213 explains that the present work uses Raw data for the Radar.
However, I did mention it in my future work and would soon write a paper explaining a new approach to use Radar for its application at Level Crossing using micro-Doppler signals. (line 447-451)
Second Review:
Please find the green highlighted changes in the manuscript especially the lines 221-227.
I hope you can review my paper within the given purpose of the manuscript and its novelty. I do appreciate your time to review and I hope the minor issue mentioned in your reviews are addressed in the submitted manuscript.
Round 4
Reviewer 2 Report
Hi, could you please provide both:
- Information as to the sensors you used for your experiments - camera and RADAR
- Examples of camera and radar data samples, ideally with detections annotated on them (qualitative results)
Author Response
Hi,
Thank you for taking the time to review my paper. I have updated my manuscript and answered your reviews in 260-294, where I have talked about the dataset used from Camera. Also, the lines 346-353 shows some qualitative results.
I hope the manuscript is sufficient to get published. The Radar as mentioned in previous reviews is not the primary choice of sensor neither the main interest of my manuscript. The manuscript discusses CCTV and its applicability with Deep Learning.

Round 5
Reviewer 2 Report
As I have said in previous reviews - this paper presents nothing significant to the theory/understanding of object detection with CNNs, and I cannot in good conscience recommend it for publication as it stands.
However, I do recognise that the "application" discussed in this paper is important - safety and security at level crossings are crucial.
Thus, I would feel more comfortable recommending this for publication if the "community" impact was more pronounced - two ways that I would suggest that this is achieved are:
- Release your annotated level crossing dataset to the community, and provide links in the abstract etc, AND/OR
- Present experiments over extended time-scales at an actual level-crossing site - having deployed your system.
Author Response:
Thank you for your email and suggestions. Like I mentioned in my last resubmission, I have clarified that my novelty is not bringing a "New Field of AI or Deep Learning Model" rather its Application at Level Crossing. I have strongly advocated it by comparison with traditional sensors, algorithms and demonstrate the effectiveness of my model using different models at the real site.
I appreciated your reviews and have answered all of them and the one you mentioned in your last email are being answered in my last resubmission file, where I have included qualitative along with quantitative results from real-site of Level Crossing.
The novelty lies there and the present work in my paper is sufficient to support it. Now your editor's reviews are correct in their own rights. I would as mentioned earlier appreciate it from your editors to check it from the scope of my paper; application of DL at Level Crossing.